# Genomic Investigation and Successful Containment of an Intermittent Common Source Outbreak of OXA-48-Producing *Enterobacter cloacae* Related to Hospital Shower Drains

Dennis Nurjadi,[a] Martin Scherrer,[a] Uwe Frank,[a,b,c] Nico T. Mutters,[a,c] Alexandra Heininger,[a,d] Isabel Späth,[a] Vanessa M. Eichel,[a] Jonas Jabs,[a,c] Katja Probst,[a] Carsten Müller-Tidow,[e] Juliane Brandt,[e] Klaus Heeg,[a] Sébastien Boutin[a]

[a]Department of Infectious Diseases, Medical Microbiology and Hygiene, Heidelberg University Hospital, Heidelberg, Germany
[b]Institute for Infection Prevention and Hospital Epidemiology, Medical Center, University of Freiburg, Freiburg im Breisgau, Germany
[c]Institute for Hygiene and Public Health, Bonn University Hospital, Bonn, Germany
[d]Department of Hospital Hygiene, University Medical Center Mannheim, Mannheim, Germany
[e]Department of Hematology, Oncology, and Rheumatology, University Hospital Heidelberg, Heidelberg, Germany

**ABSTRACT** The hospital environment has been reported as a source of transmission events and outbreaks of carbapenemase-producing Enterobacterales. Interconnected plumbing systems and the microbial diversity in these reservoirs pose a challenge for outbreak investigation and control. A total of 133 clinical and environmental OXA-48-producing *Enterobacter cloacae* isolates collected between 2015 and 2021 were characterized by whole-genome sequencing (WGS) to investigate a prolonged intermittent outbreak involving 41 patients in the hematological unit. A mock-shower experiment was performed to investigate the possible acquisition route. WGS indicated the hospital water environmental reservoir as the most likely source of the outbreak. The lack of diversity of the $bla_{OXA-48-like}$ harbouring plasmids was a challenge for data interpretation. The detection of $bla_{OXA-48-like}$-harboring *E. cloacae* strains in the shower area after the mock-shower experiment provided strong evidence that showering is the most likely route of acquisition. Initially, in 20 out of 38 patient rooms, wastewater traps and drains were contaminated with OXA-48-positive *E. cloacae*. Continuous decontamination using 25% acetic acid three times weekly was effective in reducing the trap/drain positivity in monthly environmental screening but not in reducing new acquisitions. However, the installation of removable custom-made shower tubs did prevent new acquisitions over a subsequent 12-month observation period. In the present study, continuous decontamination was effective in reducing the bacterial burden in the nosocomial reservoirs but was not sufficient to prevent environment-to-patient transmission in the long term. Construction interventions may be necessary for successful infection prevention and control.

**IMPORTANCE** The hospital water environment can be a reservoir for a multiward outbreak, leading to acquisitions or transmissions of multidrug-resistant organisms in a hospital setting. The majority of Gram-negative bacteria are able to build biofilms and persist in the hospital plumbing system over a long period of time. The elimination of the reservoir is essential to prevent further transmission and spread, but proposed decontamination regimens, e.g., using acetic acid, can only suppress but not fully eliminate the environmental reservoir. In this study, we demonstrated that colonization with multidrug-resistant organisms can be acquired by showering in showers with contaminated water traps and drains. A construction intervention by installing removable and autoclavable shower inserts to avoid sink contact during showering was effective in containing this outbreak and may be a viable alternative infection prevention and control measure in outbreak situations involving contaminated shower drains and water traps.

**KEYWORDS** carbapenemase, *Enterobacter cloacae*, hospital outbreak, infection control, whole-genome sequencing

Address correspondence to Dennis Nurjadi, dennis.nurjadi@uni-heidelberg.de, or Sébastien Boutin, sebastien.boutin@med.uni-heidelberg.de.

The emergence of carbapenemase-producing Enterobacterales (CPE) as multidrug-resistant nosocomial pathogens is an ongoing global concern (1). Besides therapeutic limitations, infections with these pathogens are associated with high morbidity and mortality (1). Genes encoding carbapenemases are usually located on mobile genetic elements, allowing transfer between organisms, which is a serious public health concern (2, 3).

*Enterobacter cloacae* are commonly encountered as commensals of the gastrointestinal tract (4). However, in recent years, *E. cloacae* has gained attention and relevance as a nosocomial pathogen, frequently associated with hospital outbreaks (5–7). Nosocomial pathogens can spread via multiple routes within a hospital setting. Besides acquisition of infections and colonization via contaminated products or involving health care workers, the environmental reservoir has been linked to long-term transmission events and outbreaks in the hospital setting (5, 8–11). In particular, water systems in health care facilities have been reported as a source of nosocomial infection, especially among immunocompromised patients in critical care units (10, 12, 13). Decontamination and infection control measures are often arduous and time-consuming because of the tenacity and persistence of this pathogen (14, 15).

Here, we report a long-term outbreak of carbapenemase-producing *E. cloacae* in patients in a hematological unit and present some of the challenges encountered in interpreting molecular typing data, focusing on the diversity of the environmental isolates and infection control measures to contain the outbreak.

## RESULTS

**Outbreak of *E. cloacae* complex with $bla_{OXA-48}$.** Starting in the second half of 2018, we noticed an increase in the detection of $bla_{OXA-48}$-positive *E. cloacae* isolates in several wards of the hematological inpatient department (Fig. 1a and b). Initial molecular typing by whole-genome sequencing (WGS) assigned most of the isolates to the sequence type 66 (ST66) clonal group (Fig. 1c). The lack of clear epidemiological overlap despite a close genetic relationship prompted a thorough environmental screening of all three wards, which revealed the presence of ST66 $bla_{OXA-48}$-carrying *E. cloacae* isolates in the sinks and plumbing system of nearly 50% of the patient rooms in the three inpatient wards of the department. Infection control measures were then implemented through weekly patient screenings and monthly environmental screenings. An overview of the timeline of events is shown in Fig. 2; the infection control and prevention measures are summarized in Table S1. Decontamination of the plumbing system by treatment with acetic acid three times a week reduced the number of contaminated rooms to under 10% (Fig. 1a and 3), but still more patients screened positive in the first months of 2019 (despite a negative screening result on admission). New detection of OXA-48-producing *E. cloacae* fluctuated and continued throughout 2019 and 2020 (Fig. 1a). To understand the transmission dynamics and optimize the containment strategies, extensive genomic analysis of clinical and environmental isolates of $bla_{OXA-48}$-harboring *E. cloacae* was performed.

**Genomic analysis of $bla_{OXA-48}$-positive *Enterobacter cloacae* isolates recovered between 2015 and 2020.** Whole-genome sequencing of all ($n = 50$) nonduplicate clinical (from rectal colonization and infection) $bla_{OXA-48}$-positive *E. cloacae* isolates recovered between 2015 and 2020 across all departments of the university hospital revealed a major cluster of *E. cloacae* strains belonging to the ST66 clonal group, of which the majority were detected in patients treated in the hematological wards, dating back to 2015 (Fig. 1c). Other clonal groups of $bla_{OXA-48}$-positive *E. cloacae* isolates were ST231, ST127, ST90, ST96, and two undetermined sequence types (one variant of ST66 with a missing *pyrG* gene and one closely related to ST90) (Fig. 1a). As expected, genetic alignment of the plasmid content revealed a lineage-independent high homology between all *E. cloacae* isolates harboring $bla_{OXA-48}$ plasmids (Fig. 1d). An overview of the antimicrobial resistance (AMR) genes present in the isolates is provided in Fig. S1. The presence or absence of AMR genes did not add any useful information due to the high clonality of our study isolates.

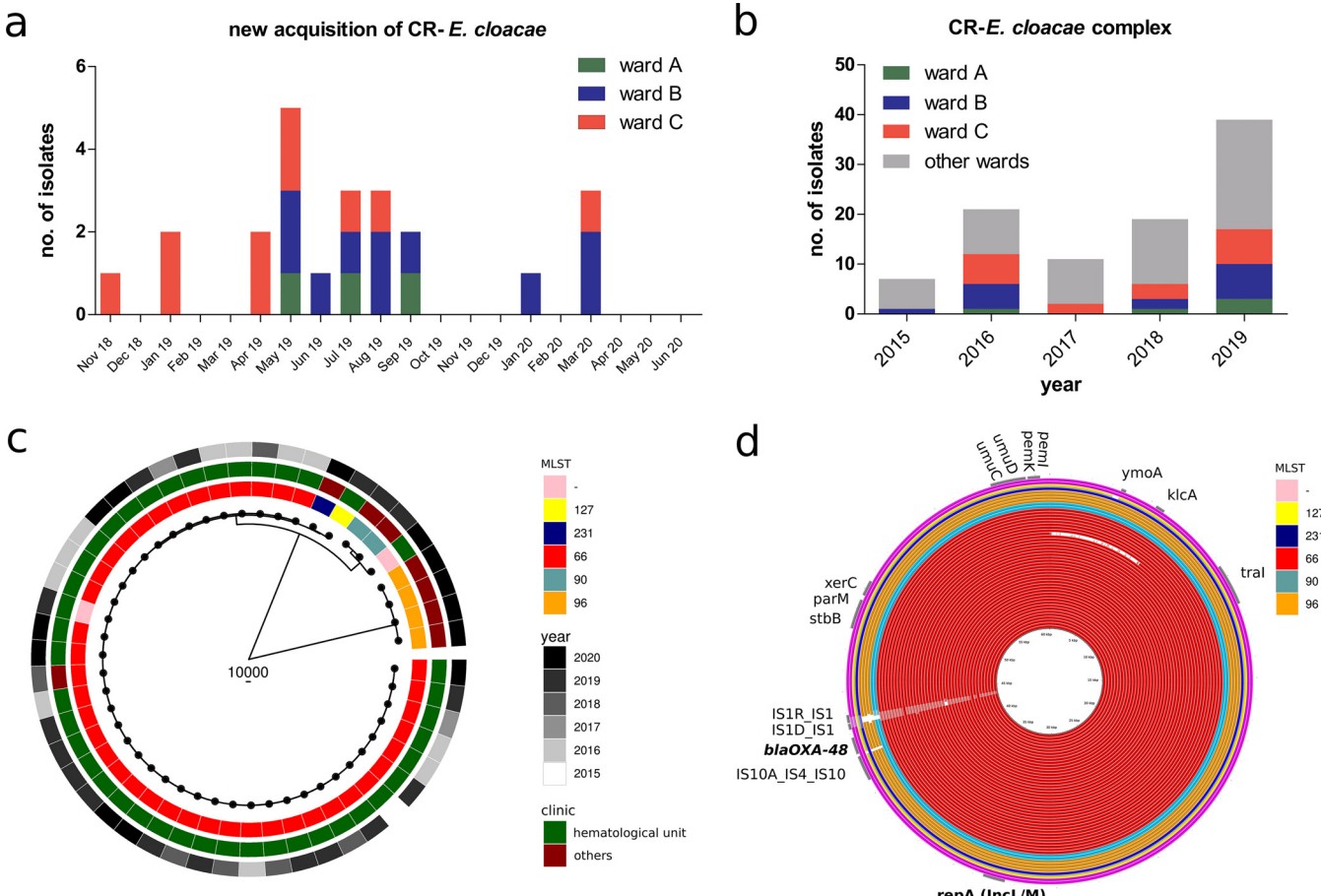

**FIG 1** Molecular characterization by whole-genome sequencing revealed an abundance of ST66 OXA-48-producing *Enterobacter cloacae* cells. (a) Overview of the accumulation of new nosocomial acquisitions of carbapenem-resistant *E. cloacae* infections/colonization in the three wards of the hematological unit. (b) Retrospective analysis of epidemiological/surveillance data revealed an overrepresentation of carbapenem-resistant (CR) *E. cloacae* in wards A, B, and C of the hematological unit. (c) Phylogeny of clinical (patient sample) *E. cloacae* isolates in the hematological unit indicated an outbreak or protracted transmission events with ST66 (multilocus sequence type, MLST) *E. cloacae* harboring a $bla_{OXA-48-like}$-containing resistance plasmid. The core genome size is 1,973 genes (2,001,303 bp), and the number of SNPs after removal of recombinant segments using Gubbins was 204,862. (d) Alignment of plasmid content from the short-read assembled draft genome sequence indicated conserved IncL/M plasmid content independent of the clonal lineage (MLST). Due to the lack of heterogeneity of $bla_{OXA-48}$-containing plasmids, analysis of plasmid genomic content is not reliable for outbreak investigations. The longest contig (60,673 bp) carrying IncL/M and $bla_{OXA-48}$ from all the different draft genomes was used as a skeleton for a mapping of all isolates carrying OXA-48 using BRIG.

Next, we constructed a minimum spanning tree (MST) based on the single nucleotide polymorphism (SNP) differences over 1,973 genes (204,862 nonrecombinant SNPs) of all sequenced clinical and environmental ST66 $bla_{OXA-48}$-positive *E. cloacae* isolates (Fig. 4). The maximum number of SNPs for all ST66 $bla_{OXA-48}$-positive *E. cloacae* isolates was 48 SNPs. Patient isolates and environmental isolates were interspersed across all branches, suggesting independent acquisition rather than patient-to-patient transmissions (Fig. 4a, left panel). By categorizing isolates by the ward in which they were first detected, we found overrepresentation of clinical and environmental isolates (Fig. 4a, middle panel). There was no direct correlation between the year of detection and genetic relatedness (Fig. 4a, right panel). Permutational multivariate analysis of variance (PERMANOVA) indicated significant ward association ($R^2 = 0.18857$, $P < 0.001$) and no significant association between patient and environment ($R^2 = 0.01657$, $P = 0.09$). An exclusive group of environmental isolates from 2020 was detected on ward B (Fig. 4a, right panel), which may be an indication of population selection due to decontamination attempts. Bayesian dating of the outbreak isolates to investigate temporal evolution was performed using BactDating (16). The root-to-tip analysis showed the strength of the temporal correlation ($R^2 = 0.2$) (Fig. S2). The time-calibrated

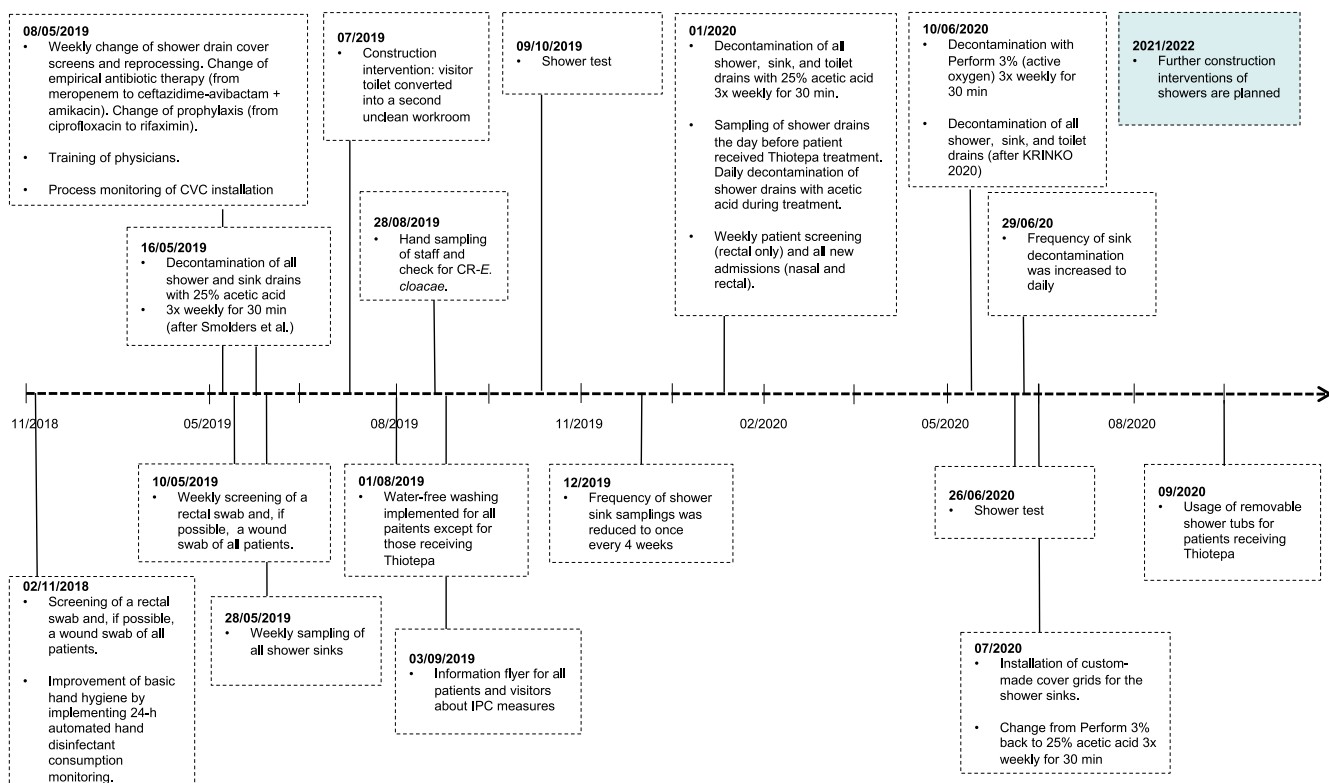

**FIG 2** Timeline of events and overview of the implementation of various infection prevention and control measures. Dates are provided in day/month/year or month/year if exact dates were not available. CR, carbapenem-resistance; CVC, central venous catheter; IPC, infection prevention and control; KRINKO, national commission for hospital hygiene and infection prevention.

phylogeny suggested a common ancestor dating back to approximately April 2009 (confidence interval, June 2005 to February 2012) (Fig. 5).

Overall, 12 of 41 patients had more than one ST66 $bla_{OXA-48}$-positive *E. cloacae* isolate over the course of their hospital stay. WGS was performed for subsequent isolates, where more than 6 months had elapsed between detections. In 67% (8/12) of these patients, the isolates were genetically closely related, with an SNP distance of ≤10. Isolates from patient 1 and patient 7 are grouped together in the MST, although these isolates were isolated in different years. Interestingly, in 4 of the 12 patients, we saw some genetic diversity in the clonal population, even among isolates recovered in the same year, with SNP distance values ranging between 1 and 40 (Fig. 4b).

**Showering as a route of acquisition.** To determine whether showering was the most likely route of acquisition, we performed a shower test in six bathrooms in which $bla_{OXA-48}$-positive *E. cloacae* had been detected in the shower drains 48 h prior to testing. In two experiments, *E. cloacae* was detected on the sedimentation plates placed on the shower tap, on a soap dish, and on the rim of a sink. In the other four experiments, we did not detect any *E. cloacae* in the sedimentation plates by culture. Other water-carrying systems such as sinks, toilets, and shower heads were screened prior to the shower tests to exclude existing reservoirs.

**Success of infection prevention measures.** Because the initial environmental screening revealed $bla_{OXA-48}$-positive *E. cloacae* in the shower drains of almost 50% of the patient rooms, decontamination of all shower and sink drains was performed three times a week with 25% acetic acid for 30 min each, according to a report by Smolders et al. (17). A summary of all infection prevention and control (IPC) measures is displayed in Table S1 and Fig. 2. At the same time, shower use was suspended, wherever possible, and replaced with water-free washing. Although these measures led to a significant reduction in sink positivity to below 10% (Fig. 3), this was not reflected immediately in a reduction of the detection of new patient cases (Fig. 1a). On the

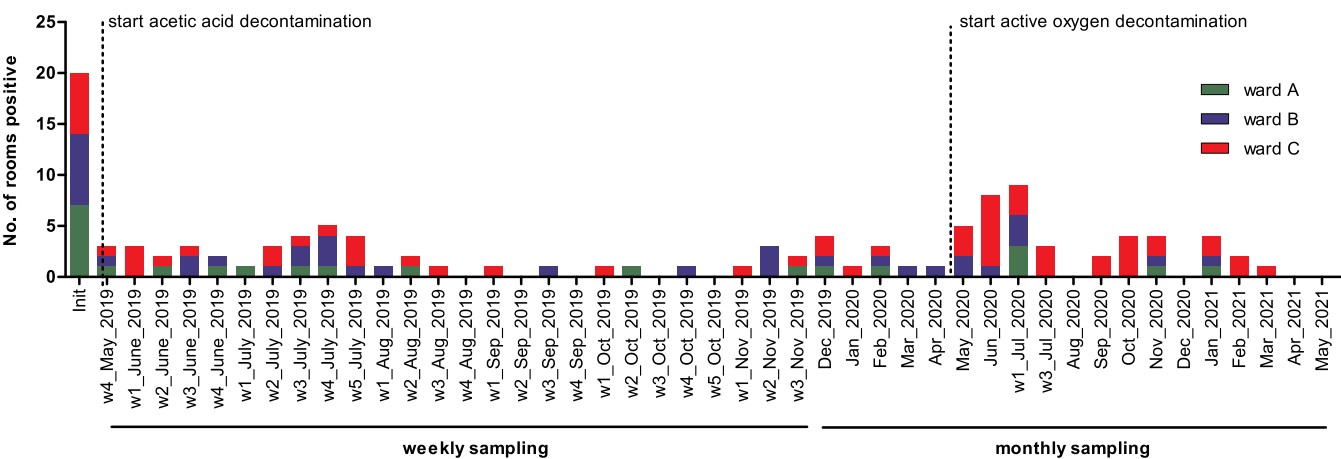

**FIG 3** Overview of environmental contamination as the acquisition source. Initial environmental screening revealed 20 patient rooms with OXA-48-producing ST66 *Enterobacter cloacae*. In the first year following the start of decontamination with acetic acid, environmental screenings were performed weekly, and monthly thereafter. The decontamination regime change to active oxygen resulted in an increase of positivity in the drainage system of the patient rooms.

recommendations of the local health authority, a new decontamination trial with Perform (active oxygen) was performed to comply with the recommendations by the Commission for Hospital Hygiene and Infection Prevention of the Robert Koch Institute (KRINKO) (18). Unfortunately, the change in the decontamination regimen led to an increase in the detection of $bla_{OXA-48}$-positive *E. cloacae* in ward C (Fig. 3). Since showering is recommended for patients receiving Thiotepa treatment (due to excretion of metabolites through the skin) (19), complete suspension of shower use was not possible. Therefore, sink decontamination was changed back to acetic acid, and removable and autoclavable shower inserts were installed additionally, to avoid backsplash by shifting the position of the shower drain and water trap (Fig. 6). As of June 2021, there had been no new acquisitions for more than 12 months.

To investigate the possible development of tolerance to the decontamination agents, five environmental isolates from 2016 (prior to decontamination with acetic acid) and five isolates from 2020 were randomly selected for susceptibility testing by broth microdilution. In general, all isolates were more susceptible to acetic acid than to Perform. There were no significant changes in MICs, so the development of tolerance by repeated exposures to acetic acid is unlikely (Fig. S3).

## DISCUSSION

Infections due to CPE in hematologic and immunocompromised patients are a major clinical concern and are associated with high morbidity and mortality (20). Although various preventive measures, such as admission screening, enforcement of basic hygiene, and contact precautions, were already in place, the acquisition and spread of multidrug-resistant Gram-negative bacilli still continued. The accumulation of OXA-48-producing *E. cloacae* was suspicious, and extensive investigations were therefore initiated. Molecular typing by WGS revealed the clonal spread between patients without an epidemiological link (patient-to-patient contact). The inclusion of environmental isolates in the analysis showed that this outbreak was related to environmental acquisition from the sinks and drains of all three affected wards as the potential sources, which highlights the importance of source control measures due to environmental reservoirs facilitating nosocomial outbreaks. Indeed, hospital sanitary facilities, including sinks, drains, and toilets, have been repeatedly identified as a potential source and reservoir for nosocomial outbreaks of multidrug-resistant Gram-negative bacilli (21–24), and the presence of an environmental reservoir has been associated with long-term outbreaks (25, 26). Over time, biofilm can form in these reservoirs, providing an ideal niche for persistence and evolution. In our study, the persistence of the original clones in the plumbing system creates a heterogeneous population of

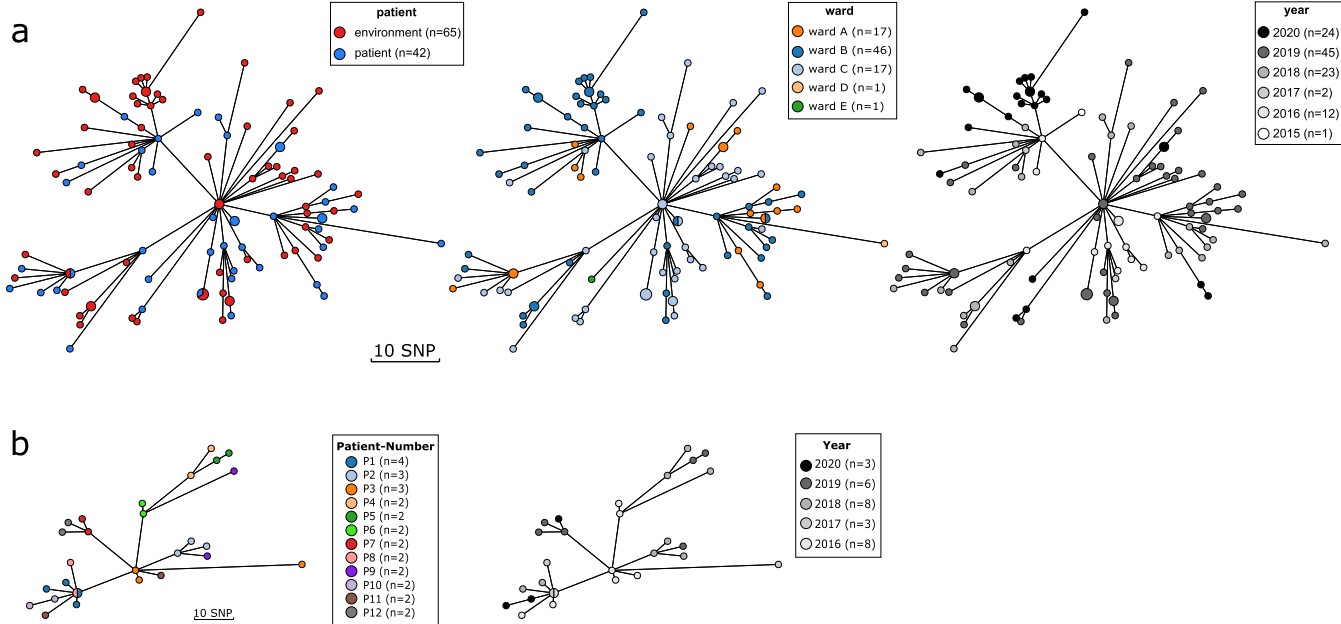

**FIG 4** Minimum spanning tree of all sequenced clinical ($n = 42$) and environmental ($n = 65$) ST66 *Enterobacter cloacae* isolates harboring $bla_{OXA-48-like}$ genes. (a) Minimum spanning tree (MST) indicated that environmental and patient isolates were genetically related (left panel). Environmental and clinical isolates from wards A, B, and C were interspersed in the MST by the ward (a, middle panel) and year (a, right panel) of acquisition, suggesting either an interconnectivity of the source of acquisition or a high transmission dynamic. The core genome size is 4,014 genes (3,892,023 bp), and the number of SNPs after removal of recombinant segments using Gubbins was 395 SNPs. (b) MST of *E. cloacae* isolates from patients with more than 2 isolates (>6 months between isolate collection). The core genome size is 4,358 genes (4,140,936 bp), and the number of SNPs after removal of recombinant segments using Gubbins was 159 SNPs.

OXA-48-producing ST66 *E. cloacae*. This heterogeneity of the reservoir was therefore blurring the transmission pattern, explaining why this outbreak remained unresolved for such a long time. Furthermore, the abundance of resistance genes and plasmids in a high-density polymicrobial environment provides ample opportunity for bacteria to interact with each other and acquire carbapenem resistance plasmids (27, 28).

Contact with contaminated water, e.g., via contaminated aerosols, is the most probable transmission route in our outbreak, as suggested by the results of the mock-shower experiment. This observation is consistent with the literature. Several studies have reported that a brief exposure and direct contact with contaminated water, along with water-related activities or aerosol inhalations, are potential transmission routes for waterborne pathogens in health care settings (29–31). Moreover, slow drainage rates and location of the drain directly underneath the water tap have been demonstrated to cause CPE present in waste traps and drains to disperse and contaminate the surrounding environment. Indeed, Aranega-Bou et al. demonstrated in an experimental setup that contaminated splashes can spread up to 1 m from the sink or tub (32).

Implementing the infection control measures for our study population was particularly challenging. The initial attempt to discontinue the use of showers and introduce water-free washing was not a viable option for all patients on these wards, since a significant portion were being treated with Thiotepa for their underlying conditions. Thiotepa, a lipophilic alkylating agent for cancer treatment, or its active metabolites can be excreted via the skin and cause cutaneous toxicities. Regular removal by showering or bathing is thus indicated (19, 33). Water-free washing may not be enough to remove the toxic residuals and prevent accumulation in the skin (19). Therefore, as a remedial action, our aim was to decontaminate the bacterial reservoir in the drainage system. Acetic acid has been proposed as an effective decontamination method for use in drains contaminated with biofilm producers (17, 34). Although the number of contaminated drains decreased significantly following the acetic acid treatment, complete eradication was not possible. A switch in the decontamination regimen from acetic acid to active oxygen-based disinfectants (Perform) was also ineffective in

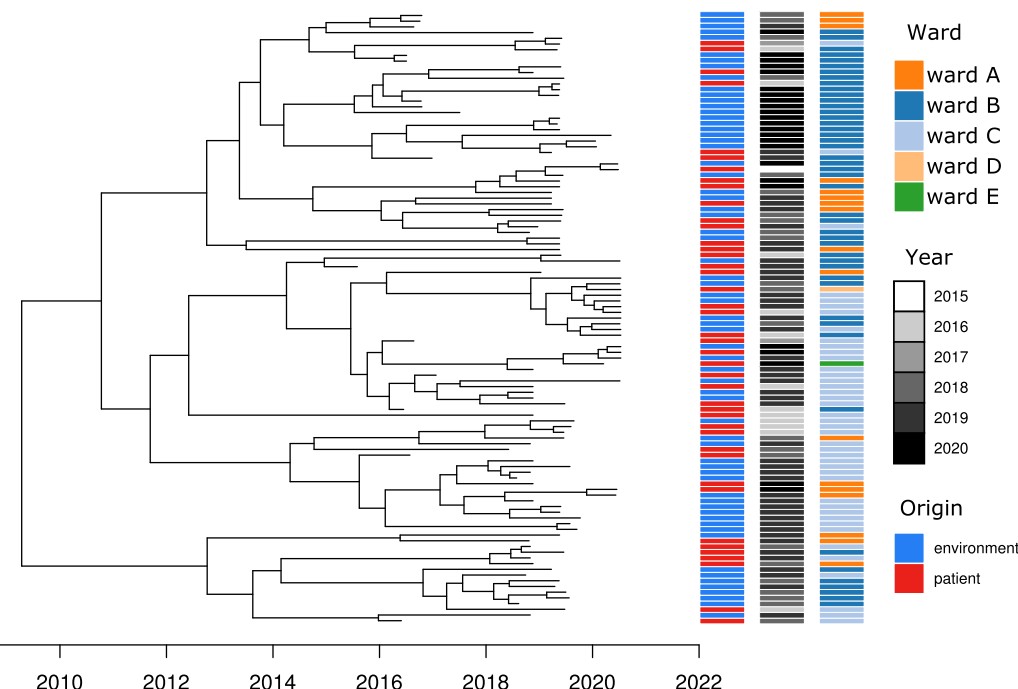

**FIG 5** Time-calibrated phylogeny of outbreak isolates. The phylogenetic tree was submitted to Bayesian dating using BactDating. The common ancestor is dated to approximately April 2009 (confidence interval, June 2005 to February 2012). The root-to-tip analysis showed that the correlation between time and phylogenetic distance is limited ($R^2 = 0.2$) (Fig. S2).

suppressing/eradicating the bacterial reservoir in the drains and led to new acquisitions. Finally, after the implementation of the removable shower inserts, which could be autoclaved regularly (once a week), the backsplash of multidrug-resistant organisms (MDRO) could be prevented effectively, and no new acquisitions were detected. Therefore, our experience indicated that showering was the main route of transmission in this outbreak.

Our study has some limitations. The interconnectedness of plumbing and the wastewater system can introduce sampling bias and complicate molecular studies investigating the evolution of strains in hospital plumbing. Although we were not able to reconstruct the exact transmission chain due to the interconnectedness of the wastewater system, multiple hospitalization periods, and complex patient movement, we demonstrate that a high-resolution typing method, such as WGS, is a useful tool for outbreak investigations and to study evolution of clinically relevant multiresistant bacteria. Furthermore, our data demonstrated that the integration of epidemiological data is essential for the interpretation of genomic data in elucidating outbreak events.

Consistent with previously published reports (17, 34), we were unable to fully decontaminate the plumbing system and had to resort to continuous suppression of the environmental reservoir using acetic acid. Nevertheless, our findings indicated no evidence of the development of tolerance to this decontaminating agent, even after repeated long-time exposure to acetic acid. Interestingly, the use of a peroxide-based disinfectant, as recommended by the National Hygiene Commission (KRINKO), was less effective than acetic acid in suppressing colonization and warrants further investigation. We demonstrated that using removable custom-made plastic shower tubs to prevent direct contact with the wastewater system of the shower area and hence prevent backsplash prevented new acquisitions in the 12-month postimplementation observation period, which is a particular strength of our study.

Taken together, the study presented here highlights and confirms the importance of hospital environmental reservoirs as a relevant source of CPE acquisition. Implementation of rigorous infection prevention and control measures, such as regular patient and

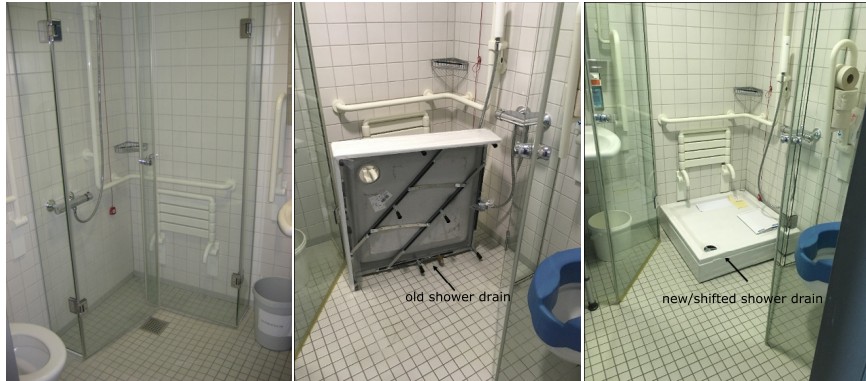

**FIG 6** Removable and autoclavable shower insert to prevent direct contact with wastewater. Following unsuccessful eradication and new detections, removable shower tubs were installed to avoid backsplash by shifting the shower drain away from the water jet from shower heads.

environmental screening procedures, can help to identify nosocomial transmission and outbreaks in a timely manner. Furthermore, a high resolution molecular typing method, such as WGS, is an important tool for outbreak investigations and is an integral component of local infection prevention and surveillance measures.

## MATERIALS AND METHODS

**Hospital and clinical setting.** This study was performed in a tertiary care hospital with over 1,900 beds in Heidelberg, Germany. The hematologic unit encompasses three wards with a total of 58 beds (ward A with 20 beds, ward B with 20 beds, and ward C with 18 beds) in 38 rooms. The patients treated in this department include those with various hematological malignancies, including immunocompromised patients who have undergone allogeneic stem cell transplantation, autologous stem cell transplantation, and treatment of leukemia and lymphoma. In addition, chimeric antigen receptor (CAR)-T-cell therapies are regularly performed. Chemotherapy treatments and autologous transplants take place on two of the three wards, while the third is an intermediate care unit specializing in allogeneic stem cell transplantation and CAR-T-cell therapy. The reporting is based on the ORION outbreak reporting guidelines (35).

**Description of study patients.** The study time frame was November 2018 to June 2020, with an additional 12-month post-outbreak observation time from July 2020 to June 2021. In total, 41 patients receiving medical treatment in the hematological department were involved in the outbreak with OXA-48-producing ST66 *E. cloacae* and were included in the analysis. All patients screened negative for carbapenem-resistant *E. cloacae* upon admission. The first detection of OXA-48-producing *E. cloacae* was in rectal screening swabs in most patients. Nine out of forty-one patients (22.0%) developed bloodstream infection with OXA-48 *E. cloacae*. One death was attributed to an *E. cloacae* bacteremia and sepsis related to the outbreak. A chronological overview of the events is provided in Fig. 2. Altogether, 133 *E. cloacae* isolates (clinical and environmental) were sequenced for a thorough investigation of this common outbreak source.

**Microbiological methods.** Admission screening for multidrug-resistant organisms (MDRO) and clinical samples were processed according to the currently valid microbiological diagnostics standard using the BD Kiestra lab automation (36). Briefly, screening samples were plated onto extended-spectrum β-lactamase (ESBL) agar (bioMérieux GmbH, Germany) and additionally onto Columbia agar with 5% sheep's blood to check for sampling validity by any bacterial growth on a nonselective universal culture medium for 24 h at 37°C. Species identification was performed using matrix-assisted laser desorption ionization–time of flight mass spectrometry (MALDI-TOF MS; Bruker Daltonics, Germany). Antibiotic susceptibility testing was performed on the Vitek 2 (bioMérieux) and interpreted according to the EUCAST clinical breakpoints of the respective year of detection. The presence of carbapenemase was detected using an in-house PCR, as described elsewhere (37, 38).

**Whole-genome sequencing and data analysis.** DNA extraction, sequencing, and data analysis were performed on the Illumina MiSeq platform, as described elsewhere (5). Library preparation was conducted using the Illumina NEXTflex rapid DNA-seq library prep kit. Sequencing was performed on a MiSeq Illumina platform (600 cycles, 2 × 300 bp). The raw sequences were controlled for quality using Sickle v1.33 (parameter: -q 30 -l 45) and assembled using SPAdes v3.13.0 (with the options –careful and –only-assembler) (39). The draft genome sequences were curated by removing contigs with a length of <1,000 bp and/or coverage of <10×. The quality of the final draft was quality controlled using QUAST v5.0.2 (40) to ensure that the final draft covered at least 95% of the expected size of the chromosome of *E. cloacae* (5.0 Mb). Annotation was performed using Prokka v1.14.1 (41). The core genome was analyzed using Roary v3.12 with only the genes present in all the isolates (42), and the SNPs were extracted from this core genome using Gubbins v3.0.0 to account for recombination events. The complete draft genome sequences were processed through available databases using ABRicate to identify the virulence factors (VFDB database), antimicrobial resistance (NCBI, CARD, ARG-ANNOT, Resfinder, and MEGARES databases), and plasmid type (PlasmidFinder database).

**Environmental sampling.** Wastewater from toilets and shower drains was collected at defined intervals, weekly during the peak of the outbreak and otherwise monthly, from all patient rooms of the three affected wards, as described elsewhere (38). The total sampling volume was approximately 40 ml, and 10 $\mu$l of the samples was cultured using chromID Carba Smart agar plates (bioMérieux GmbH) and incubated at 37°C overnight. The species was identified using MALDI-TOF MS (Bruker Daltonics). Molecular detection/confirmation of carbapenemase genes were performed using an in-house PCR described elsewhere (37).

**Mock-shower experiment.** To detect $bla_{OXA-48}$-carrying *E. cloacae* prior to the mock-shower experiment, 20 ml of liquid was collected from the waste traps and shower drains. In addition, 20 ml of fluid was collected from the shower head to exclude contamination by the shower water itself. Ten microliters of these samples was inoculated onto chromID Carba Smart agar plates (bioMérieux GmbH) and incubated for 24 h at 37°C. For the mock-shower experiment, Columbia agar supplemented with 5% sheep's blood was used as sedimentation plates. These were placed on the shower tap, the shower soap dish, the rim of the sink, the sink tap, the toilet seat lid, and the towel rack. The shower test was then performed for 10 min at a temperature of 37°C. The showering process was simulated as realistically as possible by intermittently interrupting the jet by means of the test person's hand and by varying the position of the shower head. Contamination from high splashing water was avoided by the height at which the sedimentation plates were placed and by a cover above the drain. The Columbia plates were incubated for up to 48 h at 37°C. *E. cloacae* isolates were screened for OXA-48 production using chromID Carba Smart agar plates (bioMérieux GmbH).

**Tolerance to disinfectants.** Tolerance to the decontaminating agent (acetic acid) and Perform was determined using a modified broth microdilution method based on the antibiotic susceptibility testing method in microbiological diagnostics (43). Briefly, either acetic acid (range, 0.05% to 25% vol/vol) or Perform (range, 0.20% to 100% vol/vol) was diluted 1:2 in a 96-well format in Mueller-Hinton broth, and $5 \times 10^5$ CFU was added to each well (end volume, 150 $\mu$l) and incubated overnight at 35°C $\pm$ 1°C.

**Statistical analysis.** Descriptive statistical analysis was performed using STATA 13 (StataCorp, USA) and GraphPad Prism v9.0 (GraphPad, USA). Permutational multivariate analysis of variance (PERMANOVA) was performed in R to analyze the association between the environmental isolates, patient isolates, and ward of acquisition.

**Ethics approval.** Surveillance of multidrug-resistant organisms and molecular characterization were performed in accordance with the German Infection Protection Act. The local ethical review board was consulted for the use of anonymized patient data for research purposes (S474/2018).

**Data availability.** The genome sequencing data are publicly available at NCBI GenBank under the BioProject accession number PRJNA546126. A list of accession numbers is provided in the supplementary appendix, along with the sequencing statistics of the sequenced isolates.

## SUPPLEMENTAL MATERIAL

Supplemental material is available online only.

**SUPPLEMENTAL FILE 1**, PDF file, 0.9 MB.

## ACKNOWLEDGMENTS

We thank Nicole Henny, Delal Sahin, Suzan Leccese, and Selina Hassel for excellent technical support for the WGS. We thank Deborah Lawrie-Blum for proofreading our manuscript.

All authors reviewed the manuscript prior to submission and provided consent for publication. Informed consent for the use of anonymized patient data was waived after prior consultation with the local ethical committee (Medical Faculty of the Ruprecht-Karls Universität in Heidelberg).

This study was funded by institutional funds.

We declare no conflicts of interest.

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
