## [Reviewer comments · Microbiology Spectrum]

Microbiology Spectrum

Genomic investigation and successful containment of an intermittent common source outbreak of OXA-48-producing *Enterobacter cloacae* related to hospital shower drains.

Dennis Nurjadi, Martin Scherrer, Uwe Frank, Nico Mutters, Alexandra Heining, isabel späth, vanessa eichel, jonas jabs, Katja Probst, Carsten Müller-Tidow, juliane Brandt, Klaus Heeg, and Sébastien Boutin

Corresponding Author(s): Dennis Nurjadi, Heidelberg University Hospital

Review Timeline:

Submission Date:	August 24, 2021
Editorial Decision:	October 18, 2021
Revision Received:	October 28, 2021
Accepted:	October 28, 2021

Editor: S. Wesley Long

Reviewer(s): The reviewers have opted to remain anonymous.

Transaction Report:

DOI: <https://doi.org/10.1128/Spectrum.01380-21>

October 18, 2021

Dr. Dennis Nurjadi
Heidelberg University Hospital
Department of Infectious Diseases (Medical Microbiology and Hygiene)
Im Neuenheimer Feld 324
Heidelberg
Germany

Re: Spectrum01380-21 (Genomic investigation and successful containment of an intermittent common source outbreak of OXA-48-producing *Enterobacter cloacae* related to hospital shower drains.)

Dear Dr. Dennis Nurjadi:

Thank you for submitting your manuscript to Microbiology Spectrum. When submitting the revised version of your paper, please provide (1) point-by-point responses to the issues raised by the reviewers as file type "Response to Reviewers," not in your cover letter, and (2) a PDF file that indicates the changes from the original submission (by highlighting or underlining the changes) as file type "Marked Up Manuscript - For Review Only". (3) Please make sure the genomes are released in the associated BioProject. Please provide a reviewer's link from NCBI to the submission if the submission is not yet public.

Please use this link to submit your revised manuscript - we strongly recommend that you submit your paper within the next 60 days or reach out to me. Detailed information on submitting your revised paper are below.

Link Not Available

Sincerely,

S. Wesley Long

Journals Department
Reviewer comments:

Reviewer #1 (Public repository details (Required)):

The authors state that genome assemblies are available under project number PRJNA546126. However, this project does not contain any *E. cloacae* assemblies. Additionally, uploading the sequencing reads would be more helpful for use of this data in the future. It would also be useful to include metadata (collection date, environment vs. clinical) in Table S2 as well.

Reviewer #1 (Comments for the Author):

Comments in attached review

Reviewer #2 (Comments for the Author):

In this manuscript the authors report the investigation of a OXA-48-producing *Enterobacter cloacae* outbreak involving 41 patients in a hospital hematological unit. The outbreak spanned several years and was investigated using WGS of both patients and environmental samples. The very interesting point about this paper is that once the role of showers drains in spreading the outbreak was ascertained, the authors report on the different measures that were taken to limit the spread via showers drains. I really appreciated their reporting on the very thorough efforts to limit the outbreak in the shower drains, including making controlled experiments. I would like to see more manuscripts using this approach to infection control: making experiments on the measures to be implemented, what worked and what did not. Finding outbreaks in hospital wards using WGS is now almost routine work in some parts of the world. But the real public health challenge is how to stop these outbreaks once detected. Sharing this knowledge will be helpful to public health practitioners all over the world, whether they have the chance to sequence their outbreak or not.

From this point of view this manuscript is surely a very interesting one. The methods they used to identify the environmental sources of the outbreak and the measures they took in order to clean are clearly explained. Also the bioinformatic pipeline used to analyse WGS data is clearly explained and could be replicated. It also reads very nicely.

Major concerns:

lines 225-235 239-242 & figure 4: does data have enough resolution to run BEAST or BactDating and present a dated tree? A dated tree would be easier to read and interpret and probably help to make the conclusions more formal.

327-330 The authors write: Our analysis clearly demonstrated that the close clonal relationship between patient and environmental isolates complicate the interpretation of the transmission dynamics, despite using a high-resolution typing method. However, the lack of spatio-temporal overlap among most patients, suggest that environmental reservoir is the most likely source of the outbreak.

This is not a limitation of the study, strictly speaking, it is a limitation of nature itself. Bacteria do not evolve fast enough as to be able to detect this kind of signal. Still being able to integrate genomic data with epidemiological data helps to study outbreaks, as it happened in this case. It is very important not to give false ideas about how evolution works and how much we can understand about outbreaks from genetic data. Although WGS often cannot really tell us who infected whom, it is still a very useful tool and it would be interesting to have it as an integrated tool in all CPE investigations (and resistance in general).

Minor point:

290 MDR-GN please write it long

Staff Comments:

Preparing Revision Guidelines

Please return the manuscript within 60 days; if you cannot complete the modification within this time period, please contact me. If you do not wish to modify the manuscript and prefer to submit it to another journal, please notify me of your decision immediately so that the manuscript may be formally withdrawn from consideration by Microbiology Spectrum.

Review of “Genomic investigation and successful containment of an intermittent common source outbreak of OXA-48-producing *Enterobacter cloacae* related to hospital shower drains.”

This manuscript by Nurjadi and colleagues describes a hospital-associated *Enterobacter cloacae* outbreak, including whole genome sequencing of patient and environmental samples and containment strategies of the outbreak. The results of this study highlight the role of environmental sources for outbreaks of *E. cloacae* in the hospital and demonstrate a successful containment strategy (shower inserts). The timeline of the outbreak and containment strategies are well described. However, the analyses and results from genomic and transmission analyses could be described in more detail.

Major comments:

1. The authors conclude that “Our analysis clearly demonstrated that the close clonal relationship between patient and environmental isolates complicate the interpretation of the transmission dynamics, despite using a high-resolution typing method.” However, the transmission analyses performed are not well described. There are several pieces of information missing from the manuscript.
 - a. From the methods, a core genome alignment was created using Roary including genes present in all isolates. Did this include all isolates, including those unrelated to the ST66 outbreak? How big is this core genome?
 - b. Were SNP distances derived from the core genome? SNPs that could help with transmission analyses may be present in intergenic regions or ST-specific gene content.
 - c. How were SNPs called from the alignment?
 - d. Were recombinant SNPs identified? Recombination could potentially introduce many additional SNPs into otherwise closely related genomes.
 - e. SNP distances and thresholds used to rule out transmission and the rationale for choosing these metrics are not reported.
2. The methodology used for phylogenetic analyses is not described. Additionally, phylogenies from a single cluster/ST would be more useful since genetic distances within an ST are much smaller than between STs. A time-calibrated phylogeny annotated with environment/clinical, ward, patient may be easier to interpret than the minimum spanning trees shown in Figure 4.
3. The authors state that the assemblies are available under project number PRJNA546126. However, this project does not contain any *E. cloacae* assemblies. Additionally, uploading the sequencing reads would be more helpful for use of this data in the future. It would also be useful to include metadata (collection date, environment vs. clinical) in Table S2 as well.

Minor comments

4. Line 116: “The reporting is based to” should be the “The reporting is based on”
5. Line 156: I think this should reference Table S2 instead.
6. Line 204: Table S1 and Table S2 are mixed up.
7. Lines 229-230: How was significance of clustering assessed?
8. Line 231: Molecular clock analyses are needed to assess any association between collection year and genetic distance.
9. Line 239 and Line 242: missing “difference” after SNP
10. Line 255: Should the parentheses around shower be removed?

11. There are several figures showing either timelines or number of samples/positive rooms over time.
Could any of these be combined?
12. The figure legend for Figure 2d could be more descriptive.

Point-by-point response to the reviewers

Editor's comments:

Thank you for submitting your manuscript to Microbiology Spectrum. When submitting the revised version of your paper, please provide (1) point-by-point responses to the issues raised by the reviewers as file type "Response to Reviewers," not in your cover letter, and (2) a PDF file that indicates the changes from the original submission (by highlighting or underlining the changes) as file type "Marked Up Manuscript - For Review Only". (3) Please make sure the genomes are released in the associated BioProject. Please provide a reviewer's link from NCBI to the submission if the submission is not yet public.

Response: We thank the reviewers and the handling editor for giving us the opportunity to modify and improve our manuscript. We have uploaded the genomes to NCBI. Unfortunately, we were unable to generate a reviewer's link (this option was not available and shown in the submission portal). We have, however, included a screenshot of the uploaded genomes. We have selected the option "public release" after publication, so that the genomes will be available publicly as soon as the manuscript is published. For your convenience, we have marked the responses using a different color than the original comments.

Comments of Reviewer 1:

Review of "Genomic investigation and successful containment of an intermittent common source outbreak of OXA-48-producing *Enterobacter cloacae* related to hospital shower drains." This manuscript by Nurjadi and colleagues describes a hospital-associated *Enterobacter cloacae* outbreak, including whole genome sequencing of patient and environmental samples and containment strategies of the outbreak. The results of this study highlight the role of environmental sources for outbreaks of *E. cloacae* in the hospital and demonstrate a successful containment strategy (shower inserts). The timeline of the outbreak and containment strategies are well described. However, the analyses and results from genomic and transmission analyses could be described in more detail.

Major comments:

1. The authors conclude that "Our analysis clearly demonstrated that the close clonal relationship between patient and environmental isolates complicate the interpretation of the transmission dynamics, despite using a high-resolution typing method." However, the transmission analyses performed are not well described. There are several pieces of information missing from the manuscript.

a. From the methods, a core genome alignment was created using Roary including genes present in all isolates. Did this include all isolates, including those unrelated to the ST66 outbreak? How big is this core genome?

Response: We thank reviewer #1 for the comments and apologize for the unclarity. For each figures/analyzes, we used a new core-genome related to the isolates included in the figures. The size of the core-genome is now indicated for each tree in the legend.

b. Were SNP distances derived from the core genome? SNPs that could help with transmission analyses may be present in intergenic regions or ST-specific gene content.

Response: We clarify the extraction of the SNPs in the methods section: "The core genome was analyzed using Roary 3.12 using only genes present in all the isolates (23) and the SNPs were extracted from this core genome using Gubbins 3.0.0 to account for recombination events." We agree that intergenic regions or ST-specific gene content might increase the discrimination between

STs or distant clades (figure 1). The impact is limited in our case for transmission because the SNPs are extracted from the core genome within ST66 (figure 4) and therefore the coverage of the genome is really good with >4000 genes used in the core-genome while *E. cloacae* possess around 4600 genes.

c. How were SNPs called from the alignment?

Response: See previous comment (b)

d. Were recombinant SNPs identified? Recombination could potentially introduce many additional SNPs into otherwise closely related genomes.

Response: Yes, recombinant SNPs was identified with Gubbins 3.0.0. See previous comments (c) for detail.

e. SNP distances and thresholds used to rule out transmission and the rationale for choosing these metrics are not reported.

Response: We did not used an empiric threshold such as 10 SNPs to identify identical clones but we actually showed that the maximal distance between all isolates of the ST66 was 48 SNPs which is low especially in regards of the size of the core genome (4014 genes (3892023 bp) and 395 non recombinant SNPs). Therefore, all ST66 *E. cloacae* were very closely related and considered as originating from the same clone.

2. The methodology used for phylogenetic analyses is not described. Additionally, phylogenies from a single cluster/ST would be more useful since genetic distances within an ST are much smaller than between STs. A time-calibrated phylogeny annotated with environment/clinical, ward, patient may be easier to interpret than the minimum spanning trees shown in Figure 4.

Response: See previous comments. We used Gubbins to perform the SNPs extraction and phylogeny analysis. This was not explicitly mentioned in the methods, since we referred to previously studies. We have now added this information in the methods.

As suggested by the reviewer, we also performed BactDating to time-calibrated the phylogenetic relationship of our ST66 isolate.

However, the root to tip analysis showed that the correlation time-phylogenetic distance is low ($R^2=0.2$) indicating that the strength of the temporal signal is limited . We have now added the tree as a new figure (Figure 5) in the manuscript and the root to tip analysis as a supplementary figure.

3. The authors state that the assemblies are available under project number PRJNA546126. However, this project does not contain any *E. cloacae* assemblies. Additionally, uploading the sequencing reads would be more helpful for use of this data in the future. It would also be useful to include metadata (collection date, environment vs. clinical) in Table S2 as well.

Response: We have selected the option; data will be made available upon publication. The data will be public upon publication and the metadata are already included in the NCBI Bioproject. If required, we can send the confirmation E-Mail from NCBI as a proof that the sequences have been uploaded.

Minor comments

4. Line 116: “The reporting is based to” should be the “The reporting is based on”

Response: done

5. Line 156: I think this should reference Table S2 instead.

Response: done

6. Line 204: Table S1 and Table S2 are mixed up.

Response: checked and corrected as applicable. To conform to the ASM journal requirements, we have moved the materials and methods section after the discussion.

7. Lines 229-230: How was significance of clustering assessed?

Response: We modify the term cluster for groups because it is misleading. We only visually observed a regroupment of the isolates on the MST and did not perform a clustering analysis.

8. Line 231: Molecular clock analyses are needed to assess any association between collection year and genetic distance.

Response: See previous comment (question 2)

9. Line 239 and Line 242: missing “difference” after SNP

Response: done

10. Line 255: Should the parentheses around shower be removed?

Response: done

11. There are several figures showing either timelines or number of samples/positive rooms over time. Could any of these be combined?

Response: The different figures showed different aspect/impact of the ward/time/origin on the phylogenetic distance. We could combine by changing shapes, borders or color code but we believe that separating the figures is more intuitive and readable instead of a condensed figure with overloaded data.

12. The figure legend for Figure 2d could be more descriptive.

Response : We extended the description of figure 2d to improve clarity.

Comments of Reviewer #2

Reviewer #2 (Comments for the Author):

In this manuscript the authors report the investigation of a OXA-48-producing *Enterobacter cloacae* outbreak involving 41 patients in a hospital hematological unit. The outbreak spanned several years and was investigated using WGS of both patients and environmental samples. The very interesting point about this paper is that once the role of showers drains in spreading the outbreak was ascertained, the authors report on the different measures that were taken to limit the spread via showers drains. I really appreciated their reporting on the very thorough efforts to limit the outbreak in the shower drains, including making controlled experiments. I would like to see more manuscripts using this approach to infection control: making experiments on the measures to be implemented, what worked and what did not. Finding outbreaks in hospital wards using WGS is now almost routine work in some parts of the world. But the real public health challenge is how to stop these outbreaks once detected. Sharing this knowledge will be helpful to public health practitioners all over the world, whether they have the chance to sequence their outbreak or not.

From this point of view this manuscript is surely a very interesting one. The methods they used to identify the environmental sources of the outbreak and the measures they took in order to clean are clearly explained. Also the bioinformatic pipeline used to analyse WGS data is clearly explained and could be replicated. It also reads very nicely.

Response : thank you for the positive feedback and encouraging comments. We hope to have addressed the concerns adequately.

Major concerns:

lines 225-235 239-242 & figure 4: does data have enough resolution to run BEAST or BactDating and present a dated tree? A dated tree would be easier to read and interpret and probably help to make the conclusions more formal.

Response : this was also suggested by Reviewer#1, we have now added this to the revised manuscript (new Figure 5)

327-330 The authors write: Our analysis clearly demonstrated that the close clonal relationship between patient and environmental isolates complicate the interpretation of the transmission dynamics, despite using a high-resolution typing method. However, the lack of spatio-temporal overlap among most patients, suggest that environmental reservoir is the most likely source of the outbreak.

This is not a limitation of the study, strictly speaking, it is a limitation of nature itself. Bacteria do not evolve fast enough as to be able to detect this kind of signal. Still being able to integrate genomic data with epidemiological data helps to study outbreaks, as it happened in this case. It is very important not to give false ideas about how evolution works and how much we can understand about outbreaks from genetic data. Although WGS often cannot really tell us who infected whom, it is still a very useful tool and it would be interesting to have it as an integrated tool in all CPE investigations (and resistance in general).

Response: thank you for this comment, we have removed this sentence in the limitations and rephrased to emphasize the importance of epidemiological data in interpreting WGS data.

Minor point:

290 MDR-GN please write it long

Response: done

NCBI Upload Screenshot

Submission Portal

Genome submission: SUB9937627

Carbapenem-resistant Gram Negatives Genome sequencing and assembly

✓ **BioSample: Processed** (Details)

- Successfully loaded
130 objects.
- We will automatically transform the attribute value(s) you provided as follows.
130 objects.
Download attributes file with BioSample accessions

⚙ **Genomes: Processing**

(130 genomes)

See detailed report

Download genome info file

Summary

This WGS Batch submission will be released on **2022-08-31** or upon publication, whichever is first.

Note: Release of BioProject or BioSample is also triggered by the release of linked data.

Submitter

Submitter	Dennis Nurjadi tylabor.hyg@med.uni-heidelberg.de dennis.nurjadi@med.uni-heidelberg.de
Submitting organization	Heidelberg University Hospital
Department	Medical Microbiology and Hygiene
Street	Im Neuenheimer Feld 324

City _____ Heidelberg
State/Province _____ None
Postal code _____ 69120
Country _____ Germany

BioSample general information

BioSample attributes file _____ Microbe.ncbi.xlsx (17.4 KB)

BioProject general information

BioProject ID _____ PRJNA546126

Gaps

Minimum number of N's in a row that represents a gap of estimated length 10

Type of evidence that was used to assert linkage across the assembly gaps paired-ends

Genome Info

Annotate prokaryotic genome Yes

GenBank will remove detected contamination, if possible Yes

Genome info file _____ Template_GenomeBatch.11700383121d.xlsx (25.8 KB)

Sample name	Files
BK29742	BK29742.fasta
BK30314	BK30314.fasta

Genome info file

Template_GenomeBatch.11700383121d.xlsx (25.8 KB)

Sample name	Files
BK29742	BK29742.fasta
BK30314	BK30314.fasta
D1389	D1389.fasta
D1970	D1970.fasta
D1999	D1999.fasta
D2026	D2026.fasta
D2170	D2170.fasta
D2223	D2223.fasta
D2224	D2224.fasta
D2226	D2226.fasta
D2230	D2230.fasta
D2242	D2242.fasta
D2243	D2243.fasta
D2249	D2249.fasta
D2250	D2250.fasta
D2251	D2251.fasta
D2252	D2252.fasta
D2253	D2253.fasta
D2264	D2264.fasta
D2361	D2361.fasta
D2362	D2362.fasta
D2363	D2363.fasta
D2364	D2364.fasta
D2365	D2365.fasta
D2366	D2366.fasta
D2367	D2367.fasta
D2368	D2368.fasta
D2369	D2369.fasta

October 28, 2021

Dr. Dennis Nurjadi
Heidelberg University Hospital
Department of Infectious Diseases (Medical Microbiology and Hygiene)
Im Neuenheimer Feld 324
Heidelberg
Germany

Re: Spectrum01380-21R1 (Genomic investigation and successful containment of an intermittent common source outbreak of OXA-48-producing *Enterobacter cloacae* related to hospital shower drains.)

Dear Dr. Dennis Nurjadi:

Your manuscript has been accepted, and I am forwarding it to the ASM Journals Department for publication. You will be notified when your proofs are ready to be viewed.

Sincerely,

S. Wesley Long
Editor, Microbiology Spectrum
